# Antisense Oligonucleotides Used to Identify Telomeric G-Quadruplexes in Metaphase Chromosomes and Fixed Cells by Fluorescence Lifetime Imaging Microscopy of *o*-BMVC Foci

**DOI:** 10.3390/molecules25184083

**Published:** 2020-09-07

**Authors:** Ting-Yuan Tseng, Shin-Ya Liu, Chiung-Lin Wang, Ta-Chau Chang

**Affiliations:** Institute of Atomic and Molecular Sciences, Academia Sinica, Taipei 106, Taiwan; homeotic@gmail.com (T.-Y.T.); just0717me@gmail.com (S.-Y.L.); q102081@gmail.com (C.-L.W.)

**Keywords:** G-quadruplex, telomeric G4 structure, antisense oligonucleotide, fluorescence lifetime imaging microscopy, *o*-BMVC foci

## Abstract

Identification of the existence of G-quadruplex (G4) structure, from a specific G-rich sequence in cells, is critical to the studies of structural biology and drug development. Accumulating evidence supports the existence of G4 structure in vivo. Particularly, time-gated fluorescence lifetime imaging microscopy (FLIM) of a G4 fluorescent probe, 3,6-bis(1-methyl-2-vinylpyridinium) carbazole diiodide (*o*-BMVC), was used to quantitatively measure the number of G4 foci, not only in different cell lines, but also in tissue biopsy. Here, circular dichroism spectra and polyacrylamide gel electrophoresis assays show that the use of antisense oligonucleotides unfolds their G4 structures in different percentages. Using antisense oligonucleotides, quantitative measurement of the number of *o*-BMVC foci in time-gated FLIM images provides a method for identifying which G4 motifs form G4 structures in fixed cells. Here, the decrease of the *o*-BMVC foci number, upon the pretreatment of antisense sequences, (CCCTAA)_3_CCCTA, in fixed cells and at the end of metaphase chromosomes, allows us to identify the formation of telomeric G4 structures from TTAGGG repeats in fixed cells.

## 1. Introduction

Telomeres, the ends of chromosomes, are essential for the integrity of eukaryotic chromosomes [1,2]. Human telomeres generally contain many tandem repeats of the TTAGGG/CCCTAA duplex together with a short 3′-overhang of G-rich single-stranded sequence, with 100–200 bases of (TTAGGG) repeats [3,4,5]. It is found that such G-rich sequences can rapidly form G-quadruplex (G4) structures via Hoogsteen hydrogen bonding in the presence of Na^+^ or K^+^ cations [6,7,8]. Of interest, the G4 formation at the 3′ end of the chromosome could inhibit telomerase activity and prevent telomeric elongation [9], suggesting that telomeric G4 structure is a therapeutic target [10,11,12]. Although G4 structures can be easily formed in vitro, the existence of G4 structures in vivo has been debated for a long time [13,14]. Schaffitzel et al. [15] used in vitro generated antibodies to show that the ciliate Stylonychia forms telomeric G4 structures in vivo. Granotier et al. [16] found a G4 ligand preferential binding to human chromosome ends to support the existence of G4 structures at telomeres of human cells. In addition to telomeres, Siddiqui-Jain et al. [17] provided direct evidence for the presence of G4 in a promoter region. They found that a G-rich sequence in the c-myc promotor region can form G4 structure, and its targeting with a small molecule can repress c-MYC transcription. Accumulating studies on the biological functions of G4 structures from other promotor regions, such as VEGF [18], c-KIT [19], BCL2 [20], KRAS [21], NPGPx [22], and WNT1 [23], in regulating multiple cellular processes, supported the existence of G4 structures in vivo. Recently, Hänsel-Hertsch et al. [24,25] found a high density of G4 structures, not only in the promoters, but also 5′UTR of transcribed genes, splicing sites, and mostly in cancer-related genes. Although there are increasing studies supporting the existence of G4 structures in vivo, it would be more manifest if one could develop methods to visualize G4 structures in vivo.

Bioimaging is an indispensable tool in visualizing cellular uptake and monitoring target/probe interaction in cells [26,27], as well as revealing biomolecular functions in vivo [28,29]. A fluorescent probe, 3,6-bis(1-methyl-4-vinylpyridinium) carbazole diiodide (BMVC), was the first molecule designed to visualize G4 structures at the ends of metaphase chromosomes using fluorescence imaging in 2004 [30], and fluorescence lifetime imaging microscopy (FLIM) in 2006 [31]. Recently, Biffi et al. [32] introduced a G4-specific antibody, BG4, to illustrate the existence of G4 structures in cells by immunofluorescence microscopy. Using BG4, they detected a greater number of G4 foci in human cancers of the liver and stomach than non-neoplastic tissue [33]. Furthermore, we found better G4 fluorescent probes, a BMVC isomer of 3,6-bis(1-methyl-2-vinylpyridinium) carbazole diiodide (*o*-BMVC) [34] and its derivative *o*-BMVC-12C-P [35], than BMVC for visualizing G4s in live HeLa cells, using time-gated FLIM. A further important finding is that much larger numbers of *o*-BMVC foci are detected in six types of fixed cancer cells than in three types of fixed normal cells in the binary images [36]. The use of *o*-BMVC in a clinical study of head and neck cancer supported that, the quantitative measurement of *o*-BMVC foci number in clinical biopsy provides a unique method for clinical screening of human cancers [36]. Convincing evidence suggested that the *o*-BMVC foci are mainly the G4 foci [36,37]. The chemical structure of *o*-BMVC is shown in Figure 1A. Shivalingam et al. [38] used FLIM to show that a triangulenium derivative, DAOTA-M2, indeed interacts with G4 structures in live cells. Very recently, Liu et al. [39] also applied FLIM imagery to detect more G4 foci in cancer cells than in normal cells, stained by a tripodal cationic fluorescent probe. The next question is whether one can use FLIM imaging method to identify which G4 motifs truly form G4 structures in cells. Previously, antisense oligonucleotides were used to investigate the unfolding of human telomeric G4 structures [40,41,42,43]. Here circular dichroism (CD) spectra and polyacrylamide gel electrophoresis (PAGE) assays show that the use of antisense sequences, (CCCTAA)_3_CCCTA (anti-HT) for its sense sequence of TAGGG(TTAGGG)_3_ (HT23) in human telomere (HT) and TTCCCCACCCTCCCCACCCTCA (anti-MYC) for its sense sequence of TGAGGGTGGGGAGGGTGGGGAA (MYC22) in the c-MYC promoter region, can unfold their G4 structures in different percentages. Further measurements of the number of *o*-BMVC foci in fixed HeLa cells and at the ends of metaphase chromosomes, upon the pretreatment of anti-HT sequences in time-gated FLIM images, allow us to specifically identify the existence of telomeric G4 structures formed by TTAGGG repeats in fixed cells.

## 2. Results

### 2.1. Unfolding of G4 Structures Induced by their Antisense Sequences In Vitro

PAGE assays and CD spectra were used to investigate whether the G4 structures of HT23 and MYC22 can be unfolded by their antisense sequences of anti-HT and anti-MYC in 100 mM K^+^ solution, respectively. The PAGE assays clearly showed the unfolding of a large percentage of HT23 G4 structures and the unfolding of a small percentage of MYC22 G4 structures after the overnight addition of their corresponding antisense sequences (Figure 1B). The difference in their unfolding results suggested that the use of antisense sequences to unfold their G4 structures is G-rich sequence dependent. Moreover, the gels post-stained by *o*-BMVC showed no discernible fluorescence from a single-stranded sequences of anti-HT and anti-MYC, as well as HT23/anti-HT duplexes and MYC22/anti-MYC duplexes, but strong fluorescence from telomeric G4 structures and MYC22 G4 structures (Figure 1C). Here, the weak fluorescence detected for the slow migration of MYC22 is due to dimeric G4 structures of MYC22, which is negligible in UV shadowing (Figure 1B). It appears that *o*-BMVC is a very sensitive G4 fluorescent probe. In addition, the CD spectra of 20 μM HT23 and 40 μM anti-HT, together with their mixture at 3 min, 30 min, 2 h, 8 h, and 20 h in 100 mM K^+^ solution, are shown in Figure 1D. Similarly, the CD spectra of 20 μM MYC22 and 40 μM anti-MYC, together with their mixture at 3 min, 30 min, 2 h, 8 h, and 20 h in 100 mM K^+^ solution, are shown in Figure 1E. Since the 290 nm CD band is a characteristic of non-parallel G4 structures of HT23 and the 265 nm CD band is a characteristic of parallel G4 structures of MYC22, we anticipate that the decrease of such CD signals can be used to approximately estimate the unfolded G4 structures after the addition of their antisense sequences as a function of time (Figure 1F). It is noted that the intensity of CD signals, at 290 nm for HT23 and 265nm for MYC22 after the addition of their antisense sequences can be attributed to their antisense oligonucleotides and mixtures. Nevertheless, consistent with the PAGE results, the 20 h addition of anti-HT can unfold a large percentage of the hybrid G4 structures of HT23, while the 20 h addition of anti-MYC can only unfold a small percentage of the parallel G4 structures of MYC22 in vitro.

### 2.2. The Number of G4 Foci in the Nucleus of Fixed HeLa Cells Reduced by Anti-HT Sequences

The use of *o*-BMVC for recognizing G4 structures is not only due to its much higher binding affinity to G4 DNA than to single-stranded and duplex DNA, as shown in Figure 1C, but also its longer fluorescent decay time upon binding to G4 structures (≥2.4 ns) than upon interaction with duplex or single-stranded DNA (<1.6 ns) [36]. Thus, the use of *o*-BMVC in the study of FLIM images allowed us to visualize G4 foci in HeLa cells [34,36]. Using the Otsu threshold method [44], the time gated FLIM images were analyzed and separated into two channels: red (decay time ≥ 2.4 ns) and green (decay time < 2.4 ns). The binary images of *o*-BMVC foci in fixed HeLa cells after overnight pretreatment of anti-MYC and anti-HT are shown in Figure 2A. The detection of the *o*-BMVC foci in the cytoplasm may be due to its interaction with RNA G4s [45,46], mitochondrial DNA G4s [35], or some unknown substrates in cells. Notably, the majority of DNA are in the nucleus. In addition, the use of DNase pretreatment significantly reduces the number of *o*-BMVC foci in the nucleus of HeLa cancer cells [36]. We quantitatively measured the number of *o*-BMVC foci in the nucleus of fixed HeLa cells after overnight pretreatments of LD12 (complementary linear duplex, 5′-GCGCAATTGCGC), T23 (single-stranded, 5′-TTTTTTTTTTTTTTTTTTTTTTT), anti-MYC, and anti-HT (Figure 2B). The average number of *o*-BMVC foci showed no appreciable difference upon the use of LD12, T23, and anti-MYC, but a marked decrease upon the use of anti-HT, suggesting that the use of anti-HT can unfold telomeric G4 structures in fixed HeLa cells. Although the use of anti-MYC sequences is invalid to verify the presence of G4 structures, formed by the G-rich sequences in the region of MYC promoter, the use of anti-HT sequences provides distinct evidence in identifying the existence of telomeric G4 structures formed by TTAGGG repeats in fixed cells.

### 2.3. The Number of Telomeric G4 Foci in Metaphase Chromosomes Reduced by Anti-HT Sequences

The binary images were applied to visualize the presence of *o*-BMVC foci in metaphase chromosomes extracted from HeLa and CL1-0 cancer cells, together with MRC-5 and BJ normal cells (Figure 3A–D). Again, the images were analyzed and separated into two channels: red (decay time ≥ 2.4 ns) and green (decay time < 2.4 ns). Of importance is that this imaging method is useful to map the location of each *o*-BMVC focus in metaphase chromosomes. Given that telomeres are the ends of chromosomes, and the *o*-BMVC foci detected at the end of chromosomes are mainly the telomeric G4 foci, we hypothesize that the use of anti-HT sequences can unfold telomeric G4 structures, resulting in the decrease of the number of *o*-BMVC foci at the ends of chromosomes. Since the binary images showed much greater numbers of *o*-BMVC foci at the ends of chromosomes extracted from cancer cells than from normal cells (Figure 3A,C), the binary images of *o*-BMVC foci in metaphase chromosomes, extracted from HeLa and CL1-0 cancer cells upon the pretreatment of anti-HT sequences, were conducted to test our hypothesis (Figure 3E,F). Quantitative analysis and calculation of the ratio of detecting at least one *o*-BMVC focus at the ends of the chromosomes in each image of cancer cells were plotted in Figure 3G. The average values of the probability of detecting at least one *o*-BMVC focus at the ends of the chromosomes were ~0.55 and ~0.40 for HeLa cells, as well as ~0.4 and ~0.25 for CL1-0 cancer cells, without and with the pretreatment of anti-HT sequences, respectively. The decrease in the number of *o*-BMVC foci at the ends of chromosomes upon the pretreatment of anti-HT sequences provides direct evidence of the existence of telomeric G4 structures formed by (TTAGGG) repeats.

Further analysis and measurement of the number of *o*-BMVC foci per chromosome extracted from MRC-5, BJ, HeLa, and CL1-0 cells, together with the pretreated HeLa and CL1-0 cells by anti-HT sequences, are summarized in Figure 3H. Our results showed much larger numbers of *o*-BMVC foci detected in metaphase chromosomes extracted from cancer cells than from normal cells. For example, the probability of detecting at least one *o*-BMVC focus in metaphase chromosomes is ~25% for MRC-5 and ~15% for BJ normal cells, but ~85% for HeLa and ~80% for CL1-0 cancer cells. In addition, the probability of finding at least five *o*-BMVC foci in metaphase chromosomes is 0% for both MRC-5 and BJ normal cells, but ~12% for HeLa, and ~8% for CL1-0 cancer cells. After the pretreatment of anti-HT sequences, the probability of detecting at least one *o*-BMVC focus in chromosomes decreases to ~70% for HeLa and to ~60% for CL1-0 cancer cells, while the probability of finding at least five *o*-BMVC foci in chromosomes decreases to ~5% for HeLa and ~2% for CL1-0 cancer cells. Here the imaging method provides an effective tool to map the *o*-BMVC foci in the metaphase chromosomes, and the use of anti-HT sequences unambiguously identifies the existence of *o*-BMVC foci at the ends of chromosomes resulting from G4 structures formed by telomeric G-rich sequences.

## 3. Discussion

Given that the overnight addition of anti-HT can unfold a large percentage of the G4 structures of HT23, but the overnight addition of anti-MYC can unfold a small percentage of the G4 structures of MYC22, we anticipate whether such difference in vitro is due to their G4 melting temperatures. We additionally studied three G4 structures, including GGGCGCGGGAGGAATTGGGCGGG (bcl2-M) originating from the middle four G-tracts of the P1 promoter in the bcl-2 gene [17,47], TAGGGAGGGTAGGGAGGGT (CMA) originating from the 5′-end of the c-MYC promoter NHE III_1_ [48,49], and a mitochondrial sequence GGCGTAGGTTTGGTCTAGGG (mt9437) located at positions 7114–7133 in the reversed mtDNA of the NC_012920 reference sequence, upon the addition of their antisense sequences in 100 mM K^+^ solution. The UV shadowing clearly showed the duplex formation of these pairs of G-rich/antisense sequences (Appendix A). In addition, the *o*-BMVC fluorescence in the post-stained gel assays suggested that a large percentage of bcl2-M G4 structures are unfolded by anti-bcl2-M, while the G4 structures of CMA and mt9437 are almost totally unfolded by their antisense sequences (Appendix A). We then measured the melting temperature (Tm) of HT23, MYC22, bcl2-M, CMA, and mt9437, in 100 mM K^+^ solution by CD melting curves (Appendix A). Notably, the Tm was 78.8 °C for CMA, 68.6 °C for bcl2-M, and 65 °C for HT23 in 100 mM K^+^ solution. Given that the gel assays stained by *o*-BMVC showed no appreciable fluorescence detected in the line of CMA/anti-CMA mixture, but weak fluorescence observed in the lines of HT23/anti-HT and bcl2-M/anti-bcl2-M mixtures, our results supported that the melting temperature is not sufficient to describe the unfolding difference of these G4 structures [50].

Furthermore, the 290 nm CD signal of mt9437 showed a rapid decrease after the addition of anti-mt9437 and no discernible difference after the 30 min addition of anti-mt9437 (Appendix A). That the 290 nm CD signals remained after the 20 h addition of anti-mt9437 sequences, is mainly due to the contributions from single-stranded anti-mt9437 and mt9437/anti-mt9437 duplexes. Particularly, the *o*-BMVC fluorescence was not detected in the line of mt9437/anti-mt9437 mixture in gel assays, implying that the G4 structures of mt9437 are totally unfolded by anti-mt9437 sequences. Here the gel assays allow us to approximately estimate the percentage of each G4 structure that remained after the overnight addition of its antisense sequence, which were roughly 80% for MYC22, 30% for bcl2-M, 5% for HT23, and 0% for mt9437 and CMA. Using such percentages of these G4 structures as the relative intensities after the 20 h addition of their antisense sequences, their unfolding curves can be reformed as shown in Appendix A. Here, we use a single exponential to approximately estimate their G4 unfolding times as ~1 min for mt9437, ~160 min for CMA, ~210 min for HT23, and ~710 min for bcl2-M. However, the 20% unfolding of MYC22 G4s after the 20 h addition of anti-MYC sequences cannot provide a reliable unfolding time. The previous study of unfolding kinetics of the telomeric G4 showed two decay times of 300–600 s and 4800–15,600 s [43]. Here, we use a single exponential to approximately estimate the G4 unfolding times as ~210 min for HT23, which is consistent with the longer decay time. In this work, the combination of gel assays and time-resolved CD spectra provides a method to estimate the G4 unfolding time after the addition of antisense sequences.

Previously, Moye et al. [51] observed the colocalization of telomerase and G4 structures of telomeres by using the G4 antibody BG4 and fluorescence in situ hybridization (FISH) for telomerase RNA (hTR) in vivo. Further study, based on treatment of the cells with a parallel-specific G4 porphyrin ligand, suggested that parallel G4s can form at human telomeres in cells [51]. Very recently, Bao et al. [52] reported that the telomeric sequences can form two hybrid-type and two-tetrad antiparallel G4s by in-cell ^19^F-NMR in living HeLa cells. It is not clear whether such different types of G4 structures are due to different experiments used in their studies; this deserves further investigation. Nonetheless, both experiments support the presence of telomeric G4 structures in cells. Although the study of G4 structures has markedly progressed over the last decade, many questions remain, such as identifying which G4 motifs form G4 structures, determining the major structure of G4 formation, and monitoring G4-protein interaction in cells.

Previously, we have introduced the time-gated FLIM method to detect G4 structures characterized by longer decay times (≥2.4 ns) of *o*-BMVC in live cells [34]. Notably, several groups found that the incubation of exogenous DNA sequences cannot enter the nucleus of live cells [53,54,55]. Thus, the study of using antisense oligonucleotides to identify specific G4 structures in the nucleus is conducted in fixed cells. Here, the binary imaging method together with the use of anti-MYC sequences was not able to identify the G4 structures formed by G-rich sequences in the region of the MYC promoter in fixed cells. This is not surprising because of the limited number of MYC22 G4s in cells. In contrast, this binary method together with anti-HT sequences provides distinct evidence to identify the G4 formation of telomeric G-rich sequences in fixed cells. The decrease of G4 foci, due to the treatment of anti-HT in fixed cells, is more likely due to many repeats of TTAGGG in the 3′ single-strand telomeric overhang. Notably, the importance of telomeres is not only the formation of G4 structures, but also the interaction with the cancer-associated enzyme telomerase [14,56,57]. Given that the protection of telomeres 1 (POT1) protein can unfold G4 structure prior to POT1 binding [42,51,58], visualizing how antisense sequences affect the protein binding to telomeric DNA could provide valuable insights into G4-protein interaction, for increasing our understanding of telomere biology.

G4 structures have received increasing attention as a potential target not only for cancer research [59,60,61] but also for neurodegenerative diseases [62,63,64]. Given that the binary image provides a means of detecting *o*-BMVC foci at the level of individual chromosomes, such imaging methods can be applied to verify the G4 structures formed by the pathogenic expansion of GGGGCC repeats in the C9orf72 gene in amyotrophic dementia and frontotemporal dementia at chromosome 9 [62,63,64]. The importance of this work is to illustrate that the number of *o*-BMVC foci detected in time-gate FLIM images can act as a marker to monitor G4 formation and dissociation in cells. In addition, the use of antisense sequences to unfold their corresponding G4 structures provides an effective approach, not only as an indicator for identifying the G4 structures formed by specific G4 motif, but also as a competitor for giving more information on the G4 structural biology at a molecular level in cells. Another implication of this work is to provide a method which offers a different approach for validating the exciting findings of G4s, by visualizing those G4 structures and G4–protein interactions that have been reported in cells. Such studies may unveil new insights into the biological roles of G4s and crucial information for therapeutic application, particularly into those G4s associated with cancer or other diseases.

## 4. Materials and Methods

### 4.1. DNA Preparation

DNA oligonucleotides were purchased from Bio Basic (Ontario, Canada), and dissolved in 10 mM Tris (pH 7.5). They were then subjected to heat-denaturation at 95 °C for 10 min and annealed to room temperature at a rate of 1 °C/min. The annealed oligonucleotides were stored at 4 °C overnight until further experiment. The DNA concentrations were determined using a UV-Vis absorption spectrometer (Implen, Germany).

### 4.2. Circular Dichroism (CD)

CD experiments were conducted using a spectropolarimeter (J-815, Jasco, Japan) with a bandwidth of 2 nm, at a scan speed of 50 nm/min, and a step resolution of 0.2 nm over a spectral range of 210–350 nm. The sense DNA concentration of each sample was 20 μM dissolved in 10 mM Tris (pH 7.5), and a stock solution of 3 M KCl (Sigma-Aldrich, USA) was added to the DNA samples, to attain a final K^+^ concentration. The observed signals were baseline subtracted. The melting curves were recorded at 265 nm or 290 nm, from 20 to 95 °C, with a temperature ramping rate of 1 °C/min rate controlled by a Peltier thermal coupler chamber (PFD-425S/15, Jasco, Japan).

### 4.3. Polyacrylamide Gel Electrophoresis (PAGE)

PAGE was conducted using 16% polyacrylamide and 0.5× TBE gels in the presence of 20 mM K^+^. The sense DNA concentration of each sample was 40 μM. PAGE was conducted at 175 V for 225 min at 10 °C. Gels were then photographed under ultraviolet (UV) light at 254 nm, using a digital camera. The post-stained gels were stained with 2 μM *o*-BMVC in ddH_2_O for 5 min at room temperature.

### 4.4. Cell Cultures

CL1-0, a human lung carcinoma cancer cell line, was kindly provided by Prof. P. C. Yang (National Taiwan University). The human normal lung fibroblast cell line MRC-5, human normal foreskin fibroblast cell line BJ, and human cervical adenocarcinoma cell line HeLa were obtained from the American Type Culture Collection (ATCC). CL1-0 cells were cultured in RPMI1640 medium supplemented with 10% fetal bovine serum (FBS) and 1% antibiotics. HeLa, MRC-5, and BJ cells were cultured in MEM medium supplemented with 10% FBS and 1% antibiotics. All cell lines were cultured in 5% CO_2_ at 37 °C. The antibiotic concentration was 100 U/mL penicillin and streptomycin.

### 4.5. Fluorescence Lifetime Imaging Microscopy (FLIM)

The setup of the FLIM system consisted of a picosecond diode laser (laser power, 5 mW), with an emission wavelength of 470 nm (LDH470; PicoQuant, Germany) and a ~70 ps pulse width, for the excitation of *o*-BMVC under a scanning microscope (IX-71 and FV-300; Olympus, Japan). The fluorescent signal from the *o*-BMVC was collected using a 60 × NA = 1.42 oil-immersion objective (PlanApoN; Olympus, Japan), passing through a 550/88 nm bandpass filter (Semrock, USA), followed by detection using a SPAD (PD-100-CTC; Micro Photon Devices, Italy). The fluorescence lifetime was recorded and analyzed using a time-correlated single-photon counting (TCSPC) module and software (PicoHarp 300 and SymPhoTime v5.3.2; PicoQuant, Germany). FLIM images were constructed from pixel-by-pixel lifetime information. For the study of *o*-BMVC foci in fixed HeLa cells, cells were fixed with 70% ethanol for 10 min, then 10 µM anti-MYC and anti-HT were used for pretreatment of HeLa cells overnight. After washing twice with PBS, samples were stained with 5 µM *o*-BMVC for 10 min at room temperature. For the study of *o*-BMVC foci at the ends of metaphase chromosomes, 10 µM anti-HT were used for pretreatment of samples for 2 h and stained with 5 µM *o*-BMVC for 10 min at room temperature. Quantitative analysis of *o*-BMVC foci, by using the Otsu algorithm for the image analysis, was described previously [36,37].

### 4.6. Preparation of Metaphase Chromosome Spreads

Cells were first cultured in a 10 cm dish with MEM (HeLa, MRC-5, BJ) and RPMI1640 (CL1-0) containing 10% FBS and 1% antibiotics at 37 °C in 5% CO_2_ incubator. To keep cells at metaphase of the cell cycle, cells were incubated with 10 μg/mL Demecolcine (D1925, Sigma-Aldrich, USA) for 2 h. The treated cells were re-suspended in pre-warmed 0.075 M KCl buffer for 20 min and then fixed with methanol/acetic acid (3:1) on ice. An aliquot of 20 μL of cell suspension was dropped onto a humidified slide, and then the slide put on a hot plate to evaporate the fixatives.

## 5. Conclusions

In this work, we used CD and PAGE to evaluate the unfolding capabilities of G4 structures by their antisense oligonucleotides. An *o*-BMVC molecule is an invaluable G4 fluorescence probe, which is very sensitive in detecting G4 structures, as shown in the post-stained gels. The G4 structures, unfolded in different percentages among HT23, MYC22, bcl2-M, CMA, and mt9437 by using their antisense sequences, cannot be well described by their melting temperatures, which deserves further study. In addition, the detection of the number of *o*-BMVC foci at the ends of metaphase chromosomes markedly decreased upon the treatment of anti-HT sequences in time-gated FLIM images, providing direct evidence identifying the existence of G4 structures formed by human telomeric TTAGGG repeats at the ends of metaphase chromosomes. The importance of this work is to introduce a binary imaging method, together with the use of antisense oligonucleotides, for identifying which G-rich sequences can really form G4 structures in fixed cells.

## Figures and Tables

**Figure 1 molecules-25-04083-f001:**
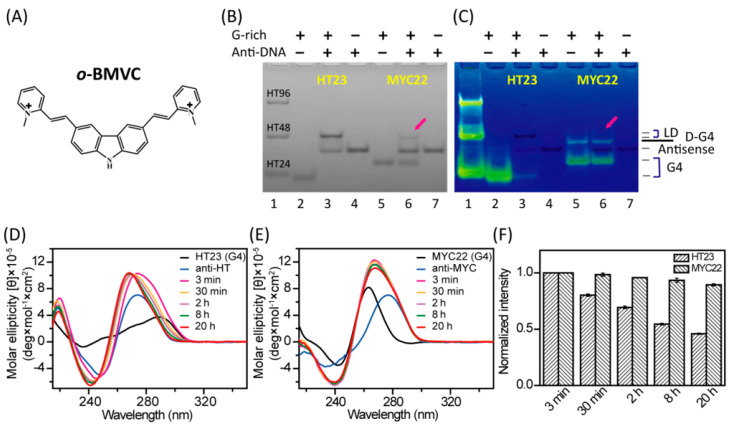
G-quadruplex (G4) structures unfolded by antisense oligonucleotides in vitro. (**A**) The chemical structure of *o*-BMVC. (**B**) Polyacrylamide gel electrophoresis (PAGE) assays of marker bands of HT24 (T_2_AG_3_)_4_, HT48 (T_2_AG_3_)_8_, and HT96 (T_2_AG_3_)_16_ (lane 1) and 40 μM HT23 G4 (lane 2), 40 μM HT23 G4 after overnight addition of 80 μM anti-HT (lane 3), 80 μM anti-HT (lane 4), 40 μM MYC22 G4 (lane 5), 40 μM MYC22 G4 after overnight addition of 80 μM anti-MYC (lane 6), and 80 μM anti-MYC (lane 7). (**C**) Post-stained PAGE assays by 2 μM *o*-BMVC in ddH_2_O for 5 min. LD and D-G4 represent linear duplex and dimeric G4 structures, respectively. (**D**) Circular dichroism (CD) spectra of 20 μM HT23 and 40 μM anti-HT together with their mixture at 3 min, 30 min, 2 h, 8 h, and 20 h, in 100 mM K^+^ solution. (**E**) CD spectra of 20 μM MYC22 and 40 μM anti-MYC together with their mixture at 3 min, 30 min, 2 h, 8 h, and 20 h, in 100 mM K^+^ solution. (**F**) The CD intensity normalized to its intensity measured right after the addition of antisense oligonucleotide at 290 nm for HT23, and at 265 nm for MYC22, as a function of time.

**Figure 2 molecules-25-04083-f002:**
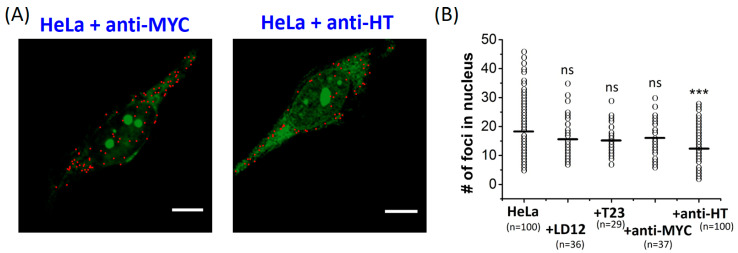
Time-gated fluorescence lifetime imaging microscopy (FLIM) imaging of *o*-BMVC foci in fixed HeLa cancer cells after the overnight pretreatment of different oligonucleotides. The binary images of *o*-BMVC foci in fixed HeLa cells after the pretreatment of anti-MYC and anti-HT (**A**). Scale bar, 10 μm. The images were presented in pseudo-color and were separated into two components, with color in red (decay time ≥ 2.4 ns) and in green (decay time < 2.4 ns). (**B**) The scatter plots of the number of *o*-BMVC foci in the nucleus of each fixed HeLa cell, without and with the pretreatment of different oligonucleotides. The average number of *o*-BMVC foci was obtained by the total number of *o*-BMVC foci in the nuclei, divided by the number of cells. (*p*-value: *** *p* < 0.001 and ns: no significance).

**Figure 3 molecules-25-04083-f003:**
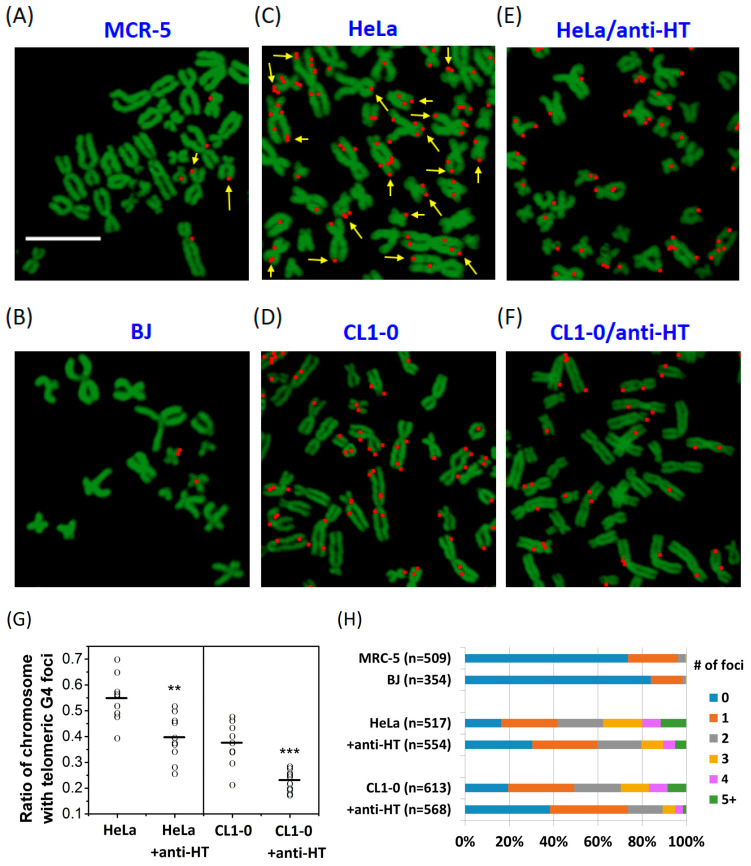
Time-gated FLIM imaging of *o*-BMVC foci on metaphase chromosomes isolated from MCR-5 (**A**) and BJ (**B**) normal cells, as well as HeLa (**C**) and CL1-0 (**D**) cancer cells, together with the anti-HT pretreated HeLa (**E**) and CL1-0 (**F**) cancer cells. Scale bar, 10 μm. The arrow symbolizes an *o*-BMVC focus located one end of the chromosome. The images were presented in pseudo-color and were separated into two components, with color in red (decay time ≥ 2.4 ns) and in green (decay time < 2.4 ns). The scale bar marked in Figurer A is also valid for Figure B–F. (**G**) The scatter plots of the ratio, detecting at least one *o*-BMVC focus at the ends of chromosomes extracted from HeLa and CL1-0 cancer cells after the addition of anti-HT sequences to the total chromosomes in each image. (*p* value: ** *p* < 0.01; *** *p* < 0.001) (**H**) The chart represents quantitative analysis of the number of *o*-BMVC foci per chromosome extracted from different cell lines.

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
