# Peer review of "Antisense Oligonucleotides Used to Identify Telomeric G-Quadruplexes in Metaphase Chromosomes and Fixed Cells by Fluorescence Lifetime Imaging Microscopy of o-BMVC Foci"

_molecules, 2020, doi:10.3390/molecules25184083_

Round 1

Reviewer 1 Report

This study consists of two parts: in vitro analysis of the rates of unfolding of various G-quadruplexes in the presence of complementary oligonucleotides, and the use of the previously-described fluorescent G4 ligand o-BMVC to probe G-quadruplex formation in fixed human cells. While the latter experiments present what may be a useful technique for examining G-quadruplex dynamics in cells, the current study does not use this technique to provide any new insights into the behaviour of G-quadruplexes, either in vitro or in vivo. Furthermore, the experiments appear way too preliminary; a substantial amount of work would need to be performed for this work to be ready for publication.

Specifically, my comments and suggestions are:

  1. For the first section regarding in vitro unfolding of G-quadruplexes in the presence of the complementary strand, the authors present this as “unfolding” by the complementary strand, but the different rates of unfolding are more likely to reflect “trapping” of the unfolded G-quadruplex by the C-strand, with the rate-limiting step being prior G-quadruplex unfolding. Indeed, this has been demonstrated by others for the telomeric G-quadruplex (Green et al., JACS 2003 (Balasubramanian lab); Li et al Biochemistry 2003 (Sugimoto lab); Chaires et al NAR 2020). This could be distinguished by determining whether the unfolding rate depends on the concentration of the complementary strand, but it is probably not worth doing this since the kinetics of unfolding of telomeric G-quadruplexes has been thoroughly studied previously, including in papers from the author’s laboratory (Li et al J Phys Chem 2014). I am not certain of the purpose of this section of the paper.
  2. It is not surprising the complementary strand to the MYC G4 does not reduce overall numbers of G4 foci in fixed cells (Fig 2B), since one would expect at most two of the foci to represent the two copies of the c-Myc promoter. I was therefore uncertain of the purpose of this experiment either.
  3. While the reductions in numbers of foci upon treatment of cells with the telomeric C-strand look promising, the data (Figs 2B and 3G) appear preliminary. To be convincing, these experiments would need to be repeated in at least 3 independent experiments, with far more cells (>100) counted in each experiment.
  4. As mentioned above, the experiments in Figs 2 and 3 present a potentially useful technique but do not in themselves provide any new biological insights. There is already a substantial amount of evidence that G-quadruplexes do form at telomeres in vivo. This technique could potentially be used to probe in vivo telomeric G4 dynamics, if the incubations were carried out for different lengths of time. One could then examine whether these dynamics were altered by manipulation of levels of proteins such as Pot1, or in response to conformation-specific G4 ligands, for example.

Author Response

Reviewer 1.

This study consists of two parts: in vitro analysis of the rates of unfolding of various G-quadruplexes in the presence of complementary oligonucleotides, and the use of the previously-described fluorescent G4 ligand o-BMVC to probe G-quadruplex formation in fixed human cells. While the latter experiments present what may be a useful technique for examining G-quadruplex dynamics in cells, the current study does not use this technique to provide any new insights into the behaviour of G-quadruplexes, either in vitro or in vivo. Furthermore, the experiments appear way too preliminary; a substantial amount of work would need to be performed for this work to be ready for publication.

--- The purpose of this work is to introduce a binary imaging tool that can be applied to identify the G4 formation from a specific sequence based on the use of its antisense oligonucleotide.

Specifically, my comments and suggestions are:

For the first section regarding in vitro unfolding of G-quadruplexes in the presence of the complementary strand, the authors present this as “unfolding” by the complementary strand, but the different rates of unfolding are more likely to reflect “trapping” of the unfolded G-quadruplex by the C-strand, with the rate-limiting step being prior G-quadruplex unfolding. Indeed, this has been demonstrated by others for the telomeric G-quadruplex (Green et al., JACS 2003 (Balasubramanian lab); Li et al Biochemistry 2003 (Sugimoto lab); Chaires et al NAR 2020). This could be distinguished by determining whether the unfolding rate depends on the concentration of the complementary strand, but it is probably not worth doing this since the kinetics of unfolding of telomeric G-quadruplexes has been thoroughly studied previously, including in papers from the author’s laboratory (Li et al J Phys Chem 2014). I am not certain of the purpose of this section of the paper.

--- Thank you. Indeed, it was suggested that the intramolecular G4 form of an oligonucleotide is in equilibrium with unfolded or partially unfolded form. Therefore, antisense oligonucleotide may function by simply “waiting” for the G4 to unfold and then binding and trapping the open form (Zaug et al Ref. 42). Here the PAGE assays showed that a large amounts of HT23 G4s are unfolded to form duplexes after the addition of its anti-HT sequences. In addition, the CMA and mt9437 G4s are likely totally unfolded to form duplexes after the addition of their antisense sequences. Thus, we named this as “unfolding” induced by the complementary strand. In addition, we agree with you that the kinetics of unfolding of telomeric G4s has been thoroughly studied previously. This is why we put the results of unfolding rates in Supplementary Information. The purpose of this section is to elucidate the possible difference of two antisense sequences with large contrast in unfolding their corresponding G4 structures between in vitro and in cells.

  1. It is not surprising the complementary strand to the MYC G4 does not reduce overall numbers of G4 foci in fixed cells (Fig 2B), since one would expect at most two of the foci to represent the two copies of the c-Myc promoter. I was therefore uncertain of the purpose of this experiment either.

--- Thank you. We agree with you that it is not surprising the anti-MYC does not reduce overall numbers of G4 foci in fixed cells. The anti-MYC experiment is used as a comparison for the anti-HT experiment.

  1. While the reductions in numbers of foci upon treatment of cells with the telomeric C-strand look promising, the data (Figs 2B and 3G) appear preliminary. To be convincing, these experiments would need to be repeated in at least 3 independent experiments, with far more cells (>100) counted in each experiment.

--- Thank you. We did these experiments at least 3 independent times. Since the importance of this study is to elucidate whether the anti-HT can reduce overall numbers of G4 foci in fixed cells, here we have studied more numbers of cells without and with the addition of anti-HT up to 100 cells as reviewer suggested (Figure 2B). The same conclusion is obtained, i.e. the anti-HT reduces overall numbers of G4 foci.

Figure 2. Time-gated FLIM imaging of o-BMVC foci in fixed HeLa cancer cells after the overnight pretreatment of different oligonucleotides. The binary images of o-BMVC foci in fixed HeLa cells after the pretreatment of anti-MYC and anti-HT (A). Scale bar, 10 μm. The images were presented in pseudocolor and were separated into two components with color in red (decay time ≥ 2.4 ns) and in green (decay time < 2.4 ns). (B) The scatter plots of the number of o-BMVC foci in the nucleus of each fixed HeLa cell without and with the pretreatment of different oligonucleotides. The average number of o-BMVC foci was obtained by the total number of o-BMVC foci in the nuclei divided by the number of cells. (P-value: ***P < 0.001 and ns: no significance)

  1. As mentioned above, the experiments in Figs 2 and 3 present a potentially useful technique but do not in themselves provide any new biological insights. There is already a substantial amount of evidence that G-quadruplexes do form at telomeres in vivo. This technique could potentially be used to probe in vivo telomeric G4 dynamics, if the incubations were carried out for different lengths of time. One could then examine whether these dynamics were altered by manipulation of levels of proteins such as Pot1, or in response to conformation-specific G4 ligands, for example.

--- Thank you for your kind suggestion. This is our hope to use such method to study in vivo telomeric G4 dynamics. Thus, collaboration is welcome.

Reviewer 2 Report

Dear Editor,

The manuscript "Antisense oligonucleotides used to identify telomeric G4 structures in metaphase chromosomes and cells by fluorescence lifetime imaging microscopy of o-BMVC foci" by Ting-Yuan Tseng and co-workers described about a potential usage of o-BMVC for visualization of G-quadruplex (G4) structure in chromosomes. The authors demonstrated that o-BMVC successfully detected G4 structure in several cell lines by the combination uses of antisense oligonucleotides targeting telomere sequences.

In my opinion, this article is well organized, and the results are described appropriately.  However, the discussion was not satisfactory, several comments should be responded before publication.

Major comments:

  1. Introduction

Although I am intimately familiar with the topic of this paper, I found some of the authors' explanations difficult to follow; why the authors examined to detect G4 by using o-BMVC? What is the advantage point of this method compared with the detection method using anti-telomere (G4) antibody? The authors should discuss about that.

  1. Figure 2 (B)   

Is this data statistically significant? The authors should do Student's t-distribution or other statistical index.

  1. Figure 2 (A) & (B)   

(1) Why the authors use MCR-5, BJ and CL1-0 cell lines? There are no explanation about those cell lines in the manuscript. The authors should indicate about that.

(2) How did the authors know those cells were in metaphase? The authors use some synchronization methods for cell preparation before LSM imaging?

  1. Figure 3 (G)   

Again, is this data statistically significant? The authors should do Student's t-distribution or other statistical index.

Minor comments:

  1. Title

Please replace “G4” to “G-quadruplex”, because “G4” is an abbreviation.

  1. Page 2, line 69  

About “anti-HT”, the name of an antisense oligonucleotide. What does “HT” stand for? The name of gene? or other?

  1. Page 4, Figure 2(A)  

Please indicate the scale bar in “HeLa + anti-HT”.

Author Response

Reviewer 2.

The manuscript "Antisense oligonucleotides used to identify telomeric G4 structures in metaphase chromosomes and cells by fluorescence lifetime imaging microscopy of o-BMVC foci" by Ting-Yuan Tseng and co-workers described about a potential usage of o-BMVC for visualization of G-quadruplex (G4) structure in chromosomes. The authors demonstrated that o-BMVC successfully detected G4 structure in several cell lines by the combination uses of antisense oligonucleotides targeting telomere sequences.

In my opinion, this article is well organized, and the results are described appropriately.  However, the discussion was not satisfactory, several comments should be responded before publication.

Major comments:

  1. Introduction

Although I am intimately familiar with the topic of this paper, I found some of the authors' explanations difficult to follow; why the authors examined to detect G4 by using o-BMVC? What is the advantage point of this method compared with the detection method using anti-telomere (G4) antibody? The authors should discuss about that.

--- Thank you. For your reference, we have previously introduced time-gated FLIM images of o-BMVC characterized by longer decay time (≥2.4 ns) to detect G4 structures in live cells (Ref. 34). The contribution of BG4 antibody to G4 research is significant (Ref. 32), however, a limitation of BG4 antibody is its restriction to fixed cells (Ref. 45). Yes, the use of antisense oligonucleotide to identify its sense G4 structure can be done in fixed cells by using BG4 antibody. The advantage of our method is that the simple addition of o-BMVC molecule to incubate with cells is more convenient than the use of BG4 antibody involving multiple processes to treat cells. According to our experience in using commercial BG4 antibody, quantitative analysis of G4 foci based on enhanced fluorescence together with its long decay time of o-BMVC upon binding to G4 structures in time-gated FLIM images provides more distinct signals than that of fluorescent dye-labeled BG4 antibody in confocal images.

  1. Figure 2 (B)

Is this data statistically significant? The authors should do Student's t-distribution or other statistical index.

--- Thank you. We have done Student’s t-test. Yes, this data is statistically significant.

 Figure 2. Time-gated FLIM imaging of o-BMVC foci in fixed HeLa cancer cells after the overnight pretreatment of different oligonucleotides. The binary images of o-BMVC foci in fixed HeLa cells after the pretreatment of anti-MYC and anti-HT (A). Scale bar, 10 μm. The images were presented in pseudocolor and were separated into two components with color in red (decay time ≥ 2.4 ns) and in green (decay time < 2.4 ns). (B) The scatter plots of the number of o-BMVC foci in the nucleus of each fixed HeLa cell without and with the pretreatment of different oligonucleotides. The average number of o-BMVC foci was obtained by the total number of o-BMVC foci in the nuclei divided by the number of cells. (P-value: ***P < 0.001 and ns: no significance)

Figure 2 (A) & (B)

  • Why the authors use MCR-5, BJ and CL1-0 cell lines? There are no explanation about those cell lines in the manuscript. The authors should indicate about that.

--- Thank you. We have rewritten Section 4.4 cell cultures as following:

CL1-0, a human lung carcinoma cancer cell line, was kindly provided by Prof. P.-C. Yang (National Taiwan University). The human normal lung fibroblast cell line MRC-5, human normal foreskin fibroblast cell line BJ, and human cervical adenocarcinoma cell line HeLa were obtained from the American Type Culture Collection (ATCC). CL1-0 cells were cultured in RPMI1640 medium supplemented with 10% fetal bovine serum (FBS) and 1% antibiotics. HeLa, MRC-5, and BJ cells were cultured in MEM medium supplemented with 10% FBS and 1% antibiotics. All cell lines were cultured in 5% CO2 at 37 °C. The antibiotic concentration was 100 U/mL penicillin and streptomycin.

  • How did the authors know those cells were in metaphase? The authors use some synchronization methods for cell preparation before LSM imaging?

--- We have added a paragraph in materials and methods to describe preparation of metaphase chromosome spreads.

4.6 Preparation of metaphase chromosome spreads

Cells were first cultured in 10 cm dish with MEM (HeLa, MRC-5, BJ) and RPMI1640 (CL1-0) containing 10% FBS and 1 % antibiotics at 37 °C in 5% CO2 incubator. To keep cells at metaphase of the cell cycle, cells were incubated with 10 μg/mL Demecolcine (D1925, Sigma-Aldrich, USA) for 2 h. The treated cells were re-suspended in pre-warmed 0.075 M KCl buffer for 20 min and then fixed with methanol/acetic acid (3:1) on ice. An aliquot of 20 μL of cell suspension was dropped on to a humidified slide and then the slide put on a hot plate to evaporate the fixatives.

 Figure 3 (G)   Again, is this data statistically significant? The authors should do Student's t-distribution or other statistical index.

--- Thank you. We have done Student’s t-test. Yes, this data is statistically significant.

Figure 3. (G) The scatter plots of the ratio detecting at least one o-BMVC focus at the ends of chromosomes extracted from HeLa and CL1-0 cancer cells after the addition of anti-HT sequences to the total chromosomes in each image. (P value: **P < 0.01; ***P < 0.001)

Minor comments:

  1. Title

Please replace “G4” to “G-quadruplex”, because “G4” is an abbreviation.

--- Thank you. We have replaced “G4 structures” by “G-quadruplexes” in the title.

  1. Page 2, line 69  

About “anti-HT”, the name of an antisense oligonucleotide. What does “HT” stand for? The name of gene? or other?

--- Thank you. HT stands for human telomere. We have revised the sentence as “Here circular dichroism (CD) spectra and polyacrylamide gel electrophoresis (PAGE) assays show that the use of antisense sequences, (CCCTAA)3CCCTA (anti-HT) for its sense sequence of TAGGG(TTAGGG)3 (HT23) in human telomere (HT) and TTCCCCACCCTCCCCACCCTCA (anti-MYC) for its sense sequence of TGAGGGTGGGGAGGGTGGGGAA (MYC22) in the c-MYC promoter region, can unfold their G4 structures in different percentages.”

 Page 4, Figure 2(A)  

Please indicate the scale bar in “HeLa + anti-HT”.

--- Thank you. We have added the scale bar in the Figure 2A of “HeLa + anti-HT”.

Reviewer 3 Report

In their study Tseng et al. have used antisense oligonucleotides to identify telomeric G4 structures  in metaphase chromosomes and cells by imaging microscopy using o-BMVC as a fluorescent G4-specific probe.

My main question is the following. The protocol used by the authors is as follows: “For the study of o-BMVC foci in fixed HeLa cells, cells were fixed with 70 % ethanol for 10 min, then 10 μM anti-MYC and anti-HT were used for pretreatment of HeLa cells overnight. After washing twice PBS, samples stained with 5 μM o-BMVC 297 for 10 min at room temperature”. The authors have detected G4 in fixed and dead cells which are not under in vivo conditions. So, in the abstract and throughout the text they cannot claim that they have provided a method for identifying which G4 motif form G4 structure in the cells. Their conclusion should be damped down.

I recognize that the study reports an interesting concept and that the data reported are sound. Only the interpretation should be recalibrated, as I stated above. However, the manuscript can be strengthened by addressing the following points:

  1. Adjust text according to the point mentioned above;
  2. 1C, post-staining with o-BMVC. The experimental protocol used to carry out the post staining is missing. In which solvent were the fluorescent probes dissolved? Was it completely soluble?.....
  3. 1D and E. The figures are not clear. There are two black CD spectra in each figure. Which is the spectrum of the G4? The annotations in the figure should be improved. Moreover the figure caption should be more precise and detailed;
  4. How were the antisense oligonucleotides delivered to the fixed cells for the overnight pretreatment? This information is missing.
  5. 2 shows that there are more o-BMVC foci in the cytoplasm than in the nucleus. Moreover it seems that there are some o-BMVC foci in the cell membrane. Please comment!
  6. What is the significance of using LD12 and T23 in Fig 2. Please explain! The text is written in a cryptic way and it is difficult to understand if essential data are missing.
  7. The author should describe in detail the protocol used to gel metaphase chromosomes;

Finally, the binary method is not able to identify G4 structures from gene promoters. This is not surprising (the number G4 structures are insufficient!), the method works only when the antisense oligonucleotide is directed against repetitive G4 motifs as telomeres and G4 expansions. This is a limit of the proposed method. However, the study provides the concept for the method: now the challenging goal will be to test the approach under real in vivo conditions.

Author Response

Reviewer 3.

In their study Tseng et al. have used antisense oligonucleotides to identify telomeric G4 structures in metaphase chromosomes and cells by imaging microscopy using o-BMVC as a fluorescent G4-specific probe.

My main question is the following. The protocol used by the authors is as follows: “For the study of o-BMVC foci in fixed HeLa cells, cells were fixed with 70 % ethanol for 10 min, then 10 μM anti-MYC and anti-HT were used for pretreatment of HeLa cells overnight. After washing twice PBS, samples stained with 5 μM o-BMVC for 10 min at room temperature”. The authors have detected G4 in fixed and dead cells which are not under in vivo conditions. So, in the abstract and throughout the text they cannot claim that they have provided a method for identifying which G4 motif form G4 structure in the cells. Their conclusion should be damped down.

--- Thank you. We have previously introduced binary FLIM method to detect G4 structures characterized by longer decay time (≥2.4 ns) of o-BMVC in live cells (Ref. 34). Notably, several groups found that the incubation of exogenous DNA sequences cannot enter the nucleus of live cells (new Ref. 52-54). This is why the study of using antisense oligonucleotides to identify specific G4 structures in the nucleus is conducted in fixed cells. For your information, confocal images showed that the Cy5-labeled DNA sequences are rarely detected in the nucleus of CL1-0 live cells.

Figure R1. Confocal images of CL1-0 live cells after the pretreatment of 15 mM Cy5-HT23 and Cy5-T24.

Here we have revised the fourth paragraph in the Discussion as following:

Previously, we have introduced time-gated FLIM method to detect G4 structures characterized by longer decay time (≥2.4 ns) of o-BMVC in live cells [34]. Notably, several groups found that the incubation of exogenous DNA sequences cannot enter the nucleus of live cells [52-54]. Thus, the study of using antisense oligonucleotides to identify specific G4 structures in the nucleus is conducted in fixed cells. Here the binary imaging method together with the use of anti-MYC sequences is not able to identify the G4 structures formed by G-rich sequences in the region of MYC promoter in fixed cells. However, this binary method together with anti-HT sequences provides a distinct evidence to identify the G4 formation of telomeric G-rich sequences in fixed cells. …

I recognize that the study reports an interesting concept and that the data reported are sound. Only the interpretation should be recalibrated, as I stated above. However, the manuscript can be strengthened by addressing the following points:

  1. Adjust text according to the point mentioned above;

--- Thank you. We have added “fixed” in those experiments conducted in fixed cells.

  1. 1C, post-staining with o-BMVC. The experimental protocol used to carry out the post staining is missing. In which solvent were the fluorescent probes dissolved? Was it completely soluble?

--- Thank you. For your information, low concentration o-BMVC (£ 10 mM) can be well dissolved in water. The post-stained gels were conducted by placing the gels in 2 mM o-BMVC in ddH2O for 5 min at room temperature.

  1. 1D and E. The figures are not clear. There are two black CD spectra in each figure. Which is the spectrum of the G4? The annotations in the figure should be improved. Moreover the figure caption should be more precise and detailed;

--- Thank you. We have revised figures and figure caption as following:

Figure 1. G4 structures unfolded by antisense oligonucleotides in vitro. A) The chemical structure of o-BMVC. B) PAGE assays of marker bands of HT24 (T2AG3)4, HT48 (T2AG3)8, and HT96 (T2AG3)16 (lane 1) and 40 mM HT23 G4 (lane 2), 40 mM HT23 G4 after overnight addition of 80 mM anti-HT (lane 3), 80 mM anti-HT (lane 4), 40 mM MYC22 G4 (lane 5), 40 mM MYC22 G4 after overnight addition of 80 mM anti-MYC (lane 6), 80 mM anti-MYC (lane 7). C) Post-stained PAGE assays by 2 mM o-BMVC in ddH2O for 5 min. LD and D-G4 represent linear duplex and dimeric G4 structures, respectively. D) CD spectra of 20 mM HT23 and 40 mM anti-HT together with their mixture at 3 min, 30 min, 2 h, 8 h, and 20 h in 100 mM K+ solution. E) CD spectra of 20 mM MYC22 and 40 mM anti-MYC together with their mixture at 3 min, 30 min, 2 h, 8 h, and 20 h in 100 mM K+ solution. F) The CD intensity normalized to its intensity measured right after the addition of antisense oligonucleotide at 290 nm for HT23 and at 265 nm for MYC22 as a function of time.

  1. How were the antisense oligonucleotides delivered to the fixed cells for the overnight pretreatment? This information is missing.

--- After fixed the cells, 10 µM antisense oligonucleotides were incubated with the cells in PBS buffer overnight. No transporter was used to deliver antisense oligonucleotides into nucleus.

  1. 2 shows that there are more o-BMVC foci in the cytoplasm than in the nucleus. Moreover it seems that there are some o-BMVC foci in the cell membrane. Please comment!

--- As we mentioned that “The detection of the o-BMVC foci in the cytoplasm may be due to its interaction with RNA G4s [45,46], mitochondrial DNA G4s [35], or some unknown substrates in cells.” Although we could study mtDNA-deficient cells to examine if the numbers of o-BMVC foci are reduced, the large deviation of the numbers of o-BMVC foci in the cytoplasm makes such study difficult. In addition, it is possible to have some o-BMVC foci in the cell membrane if o-BMVC has certain interaction with some substrates near or even the membrane. Since we have not identified those o-BMVC foci detected in the cytoplasm and in the membrane, we are not able to give a thorough discussion on those o-BMVC foci at present. Currently, we only study the o-BMVC foci in the nucleus because the use of DNAse can dramatically reduce the numbers of o-BMVC foci, implying that those o-BMVC foci in the nucleus are mainly original from DNA [36].

  1. What is the significance of using LD12 and T23 in Fig 2. Please explain! The text is written in a cryptic way and it is difficult to understand if essential data are missing.

--- We used LD12 (duplex) and T23 (single-strand) simply as control for the study of MYC22 (parallel G4) and HT23 (nonparallel G4).

  1. The author should describe in detail the protocol used to gel metaphase chromosomes;

--- We have added a paragraph in materials and methods to describe preparation of metaphase chromosome spreads.

4.6 Preparation of metaphase chromosome spreads

Cells were first cultured in 10 cm dish with MEM (HeLa, MRC-5, BJ) and RPMI1640 (CL1-0) containing 10% FBS and 1 % antibiotics at 37 °C in 5% CO2 incubator. To keep cells at metaphase of the cell cycle, cells were incubated with 10 μg/mL Demecolcine (D1925, Sigma-Aldrich, USA) for 2 h. The treated cells were re-suspended in pre-warmed 0.075 M KCl buffer for 20 min and then fixed with methanol/acetic acid (3:1) on ice. An aliquot of 20 μL of cell suspension was dropped on to a humidified slide and then the slide put on a hot plate to evaporate the fixatives.

Finally, the binary method is not able to identify G4 structures from gene promoters. This is not surprising (the number G4 structures are insufficient!), the method works only when the antisense oligonucleotide is directed against repetitive G4 motifs as telomeres and G4 expansions. This is a limit of the proposed method. However, the study provides the concept for the method: now the challenging goal will be to test the approach under real in vivo conditions.

--- Thank you. Yes, we plan to use such method to study telomeric G4 dynamics in vivo conditions.

Round 2

Reviewer 1 Report

The changes that the authors have made to the manuscript are relatively minor, and I am not entirely satisfied that all my original concerns have been adequately addressed. My reasons for this opinion are listed below, with the numbering corresponding to the 4 concerns in my original review.

  1. I am glad the authors acknowledge that the complementary strand may be functioning to trap a spontaneously-unwinding G4. However, I still maintain that the terminology used in the paper (i.e. that the antisense oligonucleotides “unfold their G4 structures”) is inappropriate, since it implies that the antisense strand is actively disrupting the G4. This might be the case, but this cannot be determined without experiments involving different concentrations of the complementary strand. This terminology should be changed throughout the paper. In addition, the concept that the different ability of each of the two sequences to trap their respective unfolded G4 is likely to be simply a reflection of the relative stabilities of the two G4s is not discussed in the manuscript, and should be added. Furthermore, some acknowledgement of previous studies measuring the unfolding rate of the telomeric G4 should be provided, and the current results compared to those papers. (A more minor point that I forgot to point out in my last review: the first two paragraphs of the Discussion would be better placed in the results section).
  2. In their response, the authors appear to have missed the main point of my concern; it’s possible I did not articulate this clearly enough. It is implied in the manuscript (by the inclusion of the MYC data in Figure 1) that the reason this technique is not valid for the MYC G4 is that the unfolding rate of the G4 is too slow, and therefore there is insufficient trapping of the unfolded DNA by the complementary strand. However, a more important and overriding consideration is that the total reduction in the number of G4 foci will never be greater than 2 foci, resulting in insufficient dynamic range to detect a reduction. Indeed, this technique is ONLY going to be useful for G4 structures that represent a large proportion of the G4 in a cell, such as telomeres; this should be made explicit. This is the reason that I think that measurement of the unfolding rate of the MYC G4 in Figure 1 is irrelevant to the rest of the paper; those data should at least be moved to the supplementary material.
  3. There has been an increase in the number of data points provided for the control and anti-HT samples, which is good. However, I requested that at least 100 cells be counted in EACH of 3 independent experiments (as is the norm in studies quantitating fluorescent foci in cells), rather than in 100 cells total across all experiments. The results of the independent experiments should ideally be presented separately to demonstrate reproducibility. This also applies to Figure 3G, which has not changed in the revised manuscript.
  4. I had missed the fact that the incubations of the complementary oligonucleotides with the cells were carried out on fixed cells, not live cells (this was picked up by the other reviewers). This means that this technique would not actually be useful for measuring the in vivo dynamics of G4 unfolding; it is essentially just a control that some of the foci detected by o-BMVC are telomeric. In my opinion, this makes it even more important that the utility of the technique is demonstrated in the current paper, through examination of a biological question.

Author Response

The changes that the authors have made to the manuscript are relatively minor, and I am not entirely satisfied that all my original concerns have been adequately addressed. My reasons for this opinion are listed below, with the numbering corresponding to the 4 concerns in my original review.

  1. I am glad the authors acknowledge that the complementary strand may be functioning to trap a spontaneously-unwinding G4. However, I still maintain that the terminology used in the paper (i.e. that the antisense oligonucleotides “unfold their G4 structures”) is inappropriate, since it implies that the antisense strand is actively disrupting the G4. This might be the case, but this cannot be determined without experiments involving different concentrations of the complementary strand. This terminology should be changed throughout the paper. In addition, the concept that the different ability of each of the two sequences to trap their respective unfolded G4 is likely to be simply a reflection of the relative stabilities of the two G4s is not discussed in the manuscript, and should be added. Furthermore, some acknowledgement of previous studies measuring the unfolding rate of the telomeric G4 should be provided, and the current results compared to those papers. (A more minor point that I forgot to point out in my last review: the first two paragraphs of the Discussion would be better placed in the results section).

--- Thank you. (1) We agree with you that “Trapping” plays an important role for the interaction of antisense oligonucleotides to their corresponding unfolded G4s. Yes, we consider that the antisense strands disrupt the equilibrium between folded and unfolded states. According to the PAGE results, the addition of 2 equivalent antisense strands changes the equilibrium significantly, only a small amounts of HT23 and almost no CMA and mt9437 G4s are detected. Thus, we named this as “unfolding” induced by the complementary strand. (2) We have measured the melting temperatures (Tm) of these G4s (Fig. S1C). As we mentioned in the first paragraph of Discussion, “Notably, the Tm is 78.8 ℃ for CMA, 68.6 ℃ for bcl2-M, and 65 ℃ for HT23 in 100 mM K+ solution. Given that the gel assays stained by o-BMVC showed no appreciable fluorescence detected in the line of CMA/anti-CMA mixture, but weak fluorescence observed in the lines of HT23/anti-HT and bcl2-M/anti-bcl2-M mixtures, our results supported that the melting temperature is not sufficient to describe the unfolding difference of these G4 structures by their antisense sequences [50].” (3) The previous study of unfolding kinetics of the telomeric G4 showed two decay times of 300-600 s and 4800-15600 s [43]. Here we use single exponential to approximately estimate the G4 unfolding times as ~210 min for HT23, which is consistent with the longer decay time. We have revised the second part of the paragraph in the Discussion as following:

… Here we use single exponential to approximately estimate their G4 unfolding times as ~1 min for mt9437, ~160 min for CMA, ~210 min for HT23, and ~710 min for bcl2-M. However, the 20% unfolding of MYC22 G4s after the 20 h addition of anti-MYC sequences cannot provide a reliable unfolding time. The previous study of unfolding kinetics of the telomeric G4 showed two decay times of 300-600 s and 4800-15600 s [43]. Here we use single exponential to approximately estimate the G4 unfolding times as ~210 min for HT23, which is consistent with the longer decay time. Here the combination of gel assays and time-resolved CD spectra provides a method to estimate the G4 unfolding time after the addition of antisense sequences.

(4) Considering the use of anti-MYC and anti-HT in fixed-cell experiments, we prefer to keep the first two paragraphs, which discuss thermal stabilities and unfolding kinetics of HT23, MYC22, bcl2-M, CMA, and mt9437 G4s in vitro in the Discussion.

  1. In their response, the authors appear to have missed the main point of my concern; it’s possible I did not articulate this clearly enough. It is implied in the manuscript (by the inclusion of the MYC data in Figure 1) that the reason this technique is not valid for the MYC G4 is that the unfolding rate of the G4 is too slow, and therefore there is insufficient trapping of the unfolded DNA by the complementary strand. However, a more important and overriding consideration is that the total reduction in the number of G4 foci will never be greater than 2 foci, resulting in insufficient dynamic range to detect a reduction. Indeed, this technique is ONLY going to be useful for G4 structures that represent a large proportion of the G4 in a cell, such as telomeres; this should be made explicit. This is the reason that I think that measurement of the unfolding rate of the MYC G4 in Figure 1 is irrelevant to the rest of the paper; those data should at least be moved to the supplementary material.

--- Thank you. We have revised the fourth paragraph in the Discussion as following:

Previously, we have introduced time-gated FLIM method to detect G4 structures characterized by longer decay time (≥2.4 ns) of o-BMVC in live cells [34]. Notably, several groups found that the incubation of exogenous DNA sequences cannot enter the nucleus of live cells [52-54]. Thus, the study of using antisense oligonucleotides to identify specific G4 structures in the nucleus is conducted in fixed cells. Here the binary imaging method together with the use of anti-MYC sequences is not able to identify the G4 structures formed by G-rich sequences in the region of MYC promoter in fixed cells. This is not surprising because of limited number of MYC22 G4s in cells. In contrast, this binary method together with anti-HT sequences provides a distinct evidence to identify the G4 formation of telomeric G-rich sequences in fixed cells. The decrease of G4 foci due to the treatment of anti-HT in fixed cells is more likely due to many repeats of TTAGGG in the 3¢ single-strand telomeric overhang. …

The anti-MYC experiments shown in Figure 1 and 2 are used as comparisons for the anti-HT experiments.

  1. There has been an increase in the number of data points provided for the control and anti-HT samples, which is good. However, I requested that at least 100 cells be counted in EACH of 3 independent experiments (as is the norm in studies quantitating fluorescent foci in cells), rather than in 100 cells total across all experiments. The results of the independent experiments should ideally be presented separately to demonstrate reproducibility. This also applies to Figure 3G, which has not changed in the revised manuscript.

--- Thank you. We have done student’s t-test. The results showed that the data are statistically significant. The reason why we add the number of data points for the control and anti-HT samples is to demonstrate that the same conclusion is obtained, i.e. the anti-HT reduces overall numbers of G4 foci.

  1. I had missed the fact that the incubations of the complementary oligonucleotides with the cells were carried out on fixed cells, not live cells (this was picked up by the other reviewers). This means that this technique would not actually be useful for measuring the in vivo dynamics of G4 unfolding; it is essentially just a control that some of the foci detected by o-BMVC are telomeric. In my opinion, this makes it even more important that the utility of the technique is demonstrated in the current paper, through examination of a biological question.

--- Thank you. We have previously introduced time-gated FLIM method to detect G4 structures characterized by longer decay time (≥2.4 ns) of o-BMVC in live cells (34). Notably, several groups found that the incubation of exogenous DNA sequences cannot enter the nucleus of live cells (52-54). This is why the study of using antisense oligonucleotides to identify specific G4 structures in the nucleus is conducted in fixed cells. However, we can use carrier to deliver them into the nucleus of live cells [35]. For your information, confocal images showed that the Cy5-labeled DNA sequences are rarely detected in the nucleus of CL1-0 live cells.

Figure R1. Confocal images of CL1-0 live cells after the pretreatment of 15 mM Cy5-HT23 and Cy5-T24.

Reviewer 2 Report

The manuscript has been revised well. I think this manuscript will be acceptable.

Author Response

Thank you for your kind suggestion.